# Therapeutic modulation of phagocytosis in glioblastoma can activate both innate and adaptive antitumour immunity

Christina A. von Roemeling[1,2,3,14], Yifan Wang [4,14], Yaqing Qie[2,5,14], Hengfeng Yuan[2,6], Hai Zhao[5], Xiujie Liu[2], Zhaogang Yang [4], Mingming Yang[4], Weiye Deng [4], Katelyn A. Bruno[7], Charles K. Chan[8], Andrew S. Lee[9,10], Stephen S. Rosenfeld[11], Kyuson Yun[12], Aaron J. Johnson[13], Duane A. Mitchell[3], Wen Jiang [4✉] & Betty Y.S. Kim [2,5✉]

Tumour cell phagocytosis by antigen presenting cells (APCs) is critical to the generation of antitumour immunity. However, cancer cells can evade phagocytosis by upregulating anti-phagocytosis molecule CD47. Here, we show that CD47 blockade alone is inefficient in stimulating glioma cell phagocytosis. However, combining CD47 blockade with temozolomide results in a significant pro-phagocytosis effect due to the latter's ability to induce endoplasmic reticulum stress response. Increased tumour cell phagocytosis subsequently enhances antigen cross-presentation and activation of cyclic GMP-AMP synthase–stimulator of interferon genes (cGAS–STING) in APCs, resulting in more efficient T cell priming. This bridging of innate and adaptive responses inhibits glioma growth, but also activates immune checkpoint. Sequential administration of an anti-PD1 antibody overcomes this potential adaptive resistance. Together, these findings reveal a dynamic relationship between innate and adaptive immune regulation in tumours and support further investigation of phagocytosis modulation as a strategy to enhance cancer immunotherapy responses.

[1] Graduate School of Biomedical Science, Mayo Clinic, Rochester, MN, USA. [2] Department of Neurosurgery, Mayo Clinic, Jacksonville, FL, USA. [3] University of Florida Brain Tumor Immunotherapy Program, Preston A. Wells Center for Brain Tumor Therapy, Lillian S. Wells Department of Neurosurgery, University of Florida, Gainesville, FL, USA. [4] Department of Radiation Oncology, The University of Texas Southwestern Medical Center, Dallas, TX, USA. [5] Department of Neurosurgery, The University of Texas MD Anderson Cancer Center, Houston, TX, USA. [6] Department of Orthopedics, Zhongshan Hospital, Fudan University, 111 Yixueyuan Road, Xuhui, Shanghai, China. [7] Department of Cardiology, Mayo Clinic, Jacksonville, FL, USA. [8] Institute for Stem Cell Biology and Regenerative Medicine, Stanford University, Stanford, CA, USA. [9] Department of Pathology, Stanford School of Medicine, Stanford, CA, USA. [10] Health Science Institute, Peking University Shenzhen, Shenzhen, China. [11] Department of Cancer Biology, Mayo Clinic, Jacksonville, FL, USA. [12] Department of Neurosurgery, Houston Methodist Research Institute, Houston, TX, USA. [13] Department of Immunology, Mayo Clinic, Rochester, MN, USA. [14] These authors contributed equally: Christina A. von Roemeling, Yifan Wang, Yaqing Qie. ✉email: wen.jiang@utsouthwestern.edu; bykim@mdanderson.org

The innate immune system acts as the first line defense to infections and malignancies[1]. The antigen presenting cells (APCs), such as monocytes, dendritic cells, and macrophages, are crucial parts of the innate immune branch that function as a bridge to the adaptive immune system. Through a process known as phagocytosis, the APCs are able to capture and eliminate transformed malignant cells, and present the tumor-derived antigens to prime T cells and activate downstream adaptive immune responses. However, tumor cells can evade phagocytosis by APCs by upregulating the "don't eat me" transmembrane protein CD47[2–4]. Blockade of CD47 has been proposed as a strategy to overcome innate immune evasion by tumor cells[5]. Administration of CD47-blocking monoclonal antibodies has been investigated in multiple preclinical human cancer models[2–7], and is capable of producing CD8+ T cell-mediated antitumor responses[8,9]. However, the nature of CD47 blockade mediated antitumor effect appears to vary across different mouse genetic backgrounds and tumor models[9–11]. Furthermore, the therapeutic effect of anti-CD47 monotherapy against solid tumors, in particular those with immune suppressive features, established in immune competent hosts remains

unclear. Here, we show that CD47 blockade alone is inefficient in stimulating glioma cell phagocytosis. Combining CD47 blockade with temozolomide (TMZ) results in a significant pro-phagocytosis effect and enhances antigen cross-presentation and activation of cGAS–STING in APCs, leading to more efficient priming of adaptive antitumour immune responses. Together, these results suggest targeting innate checkpoints that regulate phagocytosis represents a promising strategy to improve immunotherapy against immune suppressive tumors.

## Results

**Anti-CD47 treatment alone has limited anti-tumor effects**. To evaluate whether phagocytosis modulation promotes antitumour immune responses against poorly immunogenic tumors, we first evaluated whether CD47 is expressed in human and mouse glioblastoma (GBM), a malignant brain tumor. We found that CD47 is highly expressed in multiple human and murine GBM cells lines, as well as in clinical GBM tissue samples (Fig. 1a, Supplementary Fig. 1). When we treated murine glioma cells with an anti-mouse-CD47 mAb (clone MIAP410) known to functionally inhibit murine CD47–SIRPα interactions[5,12], we only

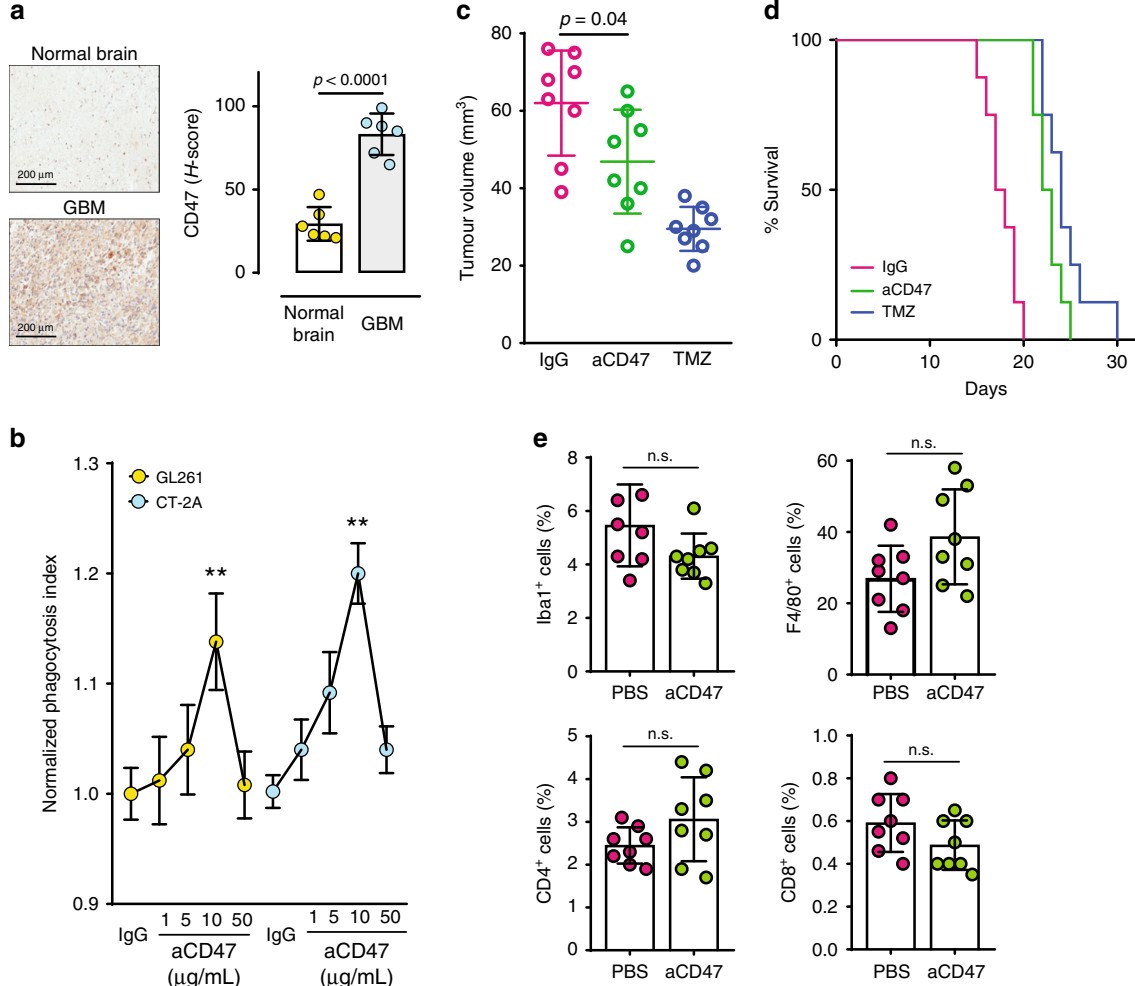

**Fig. 1 Blockade of CD47, an anti-phagocytosis molecule over-expressed in glioblastoma (GBM), does not produce significant antitumor effect. a** Immunohistochemistry analyses showing that CD47 expression is significantly elevated in GBM as compared to normal brain tissues from patient samples. Scale bar = 200 μm. n = 6/group. Unpaired student's t test. Error bar = mean ± standard deviation. **b** CD47 blockade resulted in a modest increase in murine GBM cell phagocytosis by BM phagocytes. n = 5/group. **p < 0.01 compared to IgG (unpaired Student's t test). **c** The tumor growth inhibitory effect of CD47 blockade was less than temozolomide. n = 8/group. **d** CD47 blockade did not result in improved animal survival over temozolomide treatment. n = 8/group. p = 0.1 (log-rank test). **e** No significant changes were noted in the intratumoral total (F4/80) or activated BM phagocytes (Iba1), CD4+ or CD8+ T cells after anti-CD47 antibody treatment. n = 8/group. Error bar = mean ± standard deviation. n.s. not significant.

observed a 10–20% increase in tumor cell phagocytosis by bone marrow (BM)-derived phagocytes (Fig. 1b). The increase in phagocytosis was slightly higher in human GBM cells (Supplementary Fig. 2). Similarly, when we treated orthotopically implanted syngeneic GL261 tumors with systemically administered anti-CD47 antibody, we noted a small tumor growth inhibitory effect and survival benefit (Fig. 1c, d). Tissue analyses further showed a lack of significant increase in F4/80$^+$ or Iba1$^+$ activated phagocytes, CD4$^+$ T lymphocyte and CD8$^+$ T lymphocytes for anti-CD47 monotherapy (Fig. 1e). These results suggest CD47 blockade alone is inefficient in inducing changes within the tumor immune microenvironment or eradicating murine GBMs in immune competent hosts.

**TMZ treatment induces ER-stress in GBM.** To identify strategies that can improve the effect of CD47 blockade, we next assessed whether TMZ, a DNA alkylating agent that is part of the standard of care treatment for GBM[13], can increase pro-phagocytosis signals on GBM cells. The TMZ treatment had growth inhibitory effects on the glioma cells (Supplementary Fig. 3). We found that glioma cells treated with TMZ transiently increased the translocation of the chaperone protein calreticulin, a pro-phagocytosis molecule that interacts with low density lipoprotein receptor-related protein 1 (LRP1)/CD91 receptor on phagocytes, from the ER lumen to the plasma membrane in a dose-dependent manner (Fig. 2a, b). This effect was not observed in normal BM cells (Fig. 2c). The translocation of calreticulin is associated with increased phagocytosis of tumor cells (Fig. 2d, Supplementary Fig. 4) and transcription of ER stress response-specific targets damage inducible protein 3 (DDIT3), homocysteine inducible ER protein with ubiquitin like domain 1 (HERPUD1), growth arrest and DNA damage inducible alpha (GADD45α) (Fig. 2e, Supplementary Fig. 5), as well as protein expression of binding immunoglobulin protein (BiP) and C/EBP homologous protein (CHOP) Fig. 2f, Supplementary Fig. 6)[14,15]. TMZ treatment also promoted phosphorylation of eukaryotic translation initiation factor 2A (eIF2A) at serine 51 (Fig. 2f, Supplementary Fig. 6), an event that is required for calreticulin translocation to the cell surface, consistent with the activation of the PKR-like ER kinase branch of unfolded protein response[16–18]. TMZ treatment did not induce significant apoptosis until the dose exceeded 500 μM (Supplementary Fig. 7a). We did not observe a significant increase in the release of HMGB1 after TMZ treatment at lower concentrations (50 or 100 μM) (Supplementary Fig. 7b), suggesting that the calreticulin translation likely occurred during early phases of apoptosis or within surviving cells. High concentration TMZ (800 μM) treatment induced substantial apoptosis of the THP-1 cell line (Supplementary Fig. 7c). Treatment of the glioma cells with the ER stress inhibitor 4-phenylbutyric acid (4-PBA) largely abrogated TMZ-induced calreticulin translocation and phagocytosis (Fig. 2g)[19], supporting a critical role of ER stress responses in generating these effects.

**TMZ-induced ER-stress in GBM depends on MGMT deficiency.** Given that the antitumor effect of TMZ in GBM is highly dependent on the expression of O-6-methyl-guanine methyltransferase (MGMT)[20], we next tested whether MGMT overexpression affects the ER stress-inducing effect of TMZ on glioma cells. We found that induced expression of MGMT transcriptionally downregulated *DDIT3*, which encodes CHOP, and disrupted TMZ-induced calreticulin plasma membrane translocation (Supplementary Fig. 8a–c). This effect, however, could be rescued by the addition of O$^6$-benzylguanine (O6BG), a synthetic derivative of guanine and substrate competitor for MGMT

methyltransferase activity (Supplementary Fig. 8d, e)[21]. In addition, we tested a human glioma cell line, U138, which has high endogenous levels of MGMT expression. Compared with the MGMT-deficient cell line U251, U138 was resistant to TMZ-induced ER-stress and calreticulin translocation (Supplementary Fig. 8f, g). Accordingly, TMZ treatment only slightly increased (~15%) U138 phagocytosis by THP-1, whereas U251 phagocytosis was 70% higher (Supplementary Fig. 8h). Together these results suggest that TMZ-induced genomic instability likely contributed to the ER stress responses in the MGMT-deficient glioma cells[22], leading to the translocation of calreticulin to the cell surface, where it facilitates the recognition and phagocytosis of tumor cells by phagocytes[23].

**Combo treatment increases glioma cell phagocytosis by APCs.** Since CD47 blockade and TMZ promote glioma cell phagocytosis by BM cells through distinctive mechanisms, we next investigated whether they can be combined to produce more potent phagocytic effects. We found that when TMZ was given with anti-CD47 antibody, a significantly higher degree of glioma cell phagocytosis was observed as compared to either treatment alone (Fig. 3a, b). Enhanced tumor cell phagocytosis was strongly dependent on CRT, as the addition of a CRT blocking peptide largely neutralized this effect (Fig. 3c). We tested another chemotherapy drug, cisplatin, which is used to treat glioma and can induce ER stress response[24]. We found that cisplatin treatment also induced ER-stress and calreticulin translocation in GL261 cells (Supplementary Fig. 9a). We observed increased phagocytosis by BM cells after GL261 cells were treated by cisplatin for 48 h. When combined with anti-CD47, further increase of phagocytosis was observed (Supplementary Fig. 9b). To test the prophagocytic effect of cisplatin in other solid cancers, we next investigated if the combination of cisplatin and anti-CD47 had better efficacy than each treatment used alone in breast cancer model. We found 48 h of cisplatin treatment enhanced calreticulin translocation and induced ER-stress of the murine breast adenocarcinoma E0771 cells (Supplementary Fig. 9c–e) and significantly increased BM-derived APC phagocytosis of the E0771 cells (Supplementary Fig. 9f, g). These results indicated that the improved phagocytosis effect of ER-stress inducing agents and CD47 blockade is likely to apply to other therapeutic agents beyond TMZ and to other solid tumors.

Since microglia act as the predominant antigen presenting cells in the central nervous system, we next investigated whether combined TMZ and CD47 blockade produces a similar pro-phagocytosis effect of microglia against GBM cells. Similar to BM phagocytes, the combined treatment significantly increased GBM cell phagocytosis by microglia as compared to both CD47 blockade and TMZ treatment alone (Supplementary Fig. 10). To determine which mononuclear population was more predominant in the tumor microenvironment after the combo treatment, we used the CCR2$^{RFP}$CX3CR1$^{GFP}$ reporter mice to track and distinguish BM cells and microglia within gliomas. Tissue immunofluorescence results demonstrated a large increase in both the number and depth of tissue infiltration by BM cells after the combo treatment while microglia remained largely sequestered at the tumor periphery (Fig. 3d, e). These data suggested that BM might be the predominant APC population in response to TMZ and anti-CD47 combination treatment within the glioma microenvironment.

**Combo treatment increases antigen cross-presentation.** To examine whether the increased tumor cell phagocytosis also enhanced antigen-cross presentation by these professional APCs, we cocultured cytoplasmic-ovalbumin (cOVA) expressing

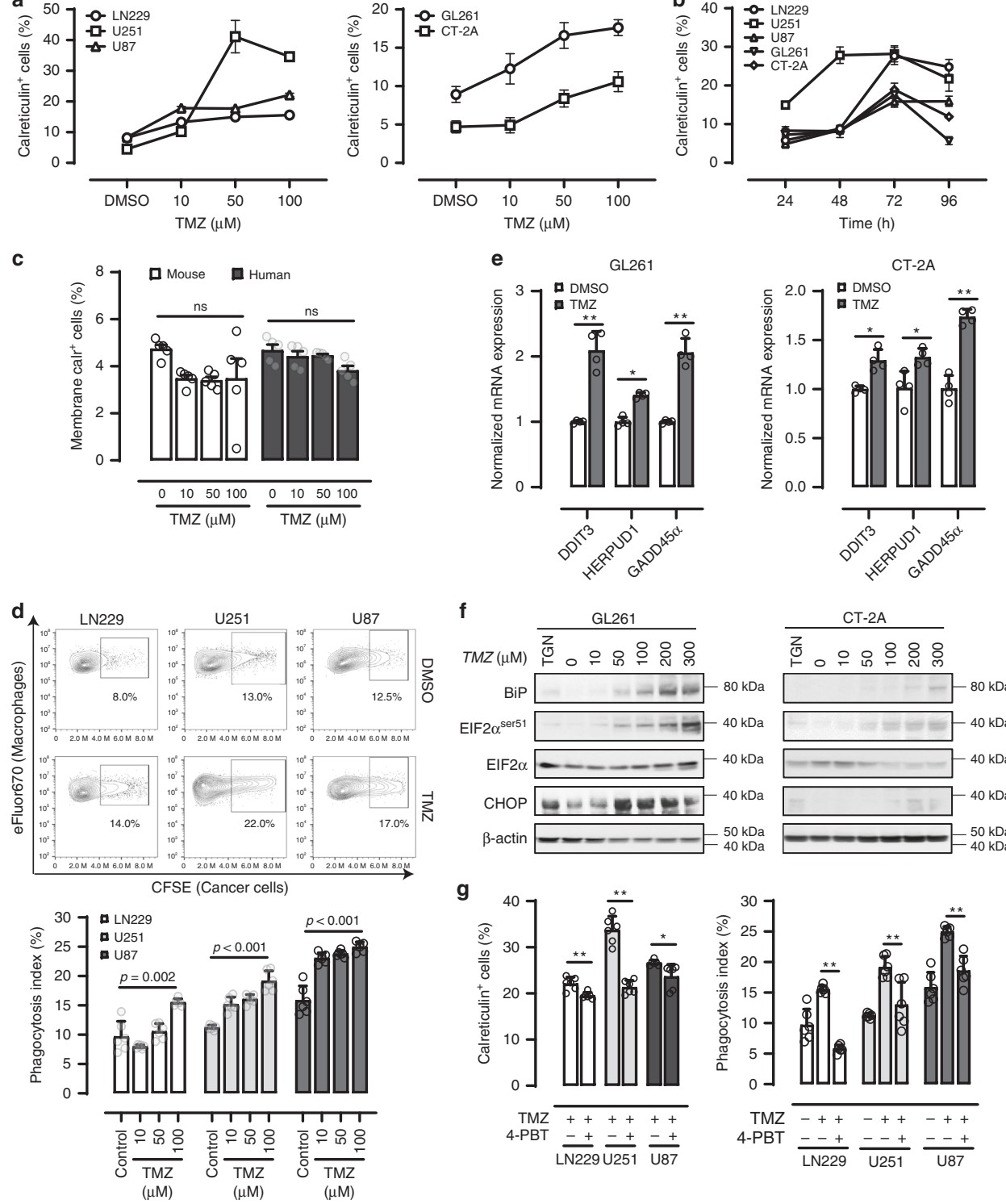

**Fig. 2 Temozolomide (TMZ) promotes GBM cell phagocytosis by bone marrow (BM)-derived phagocytes. a** TMZ induces translocation of the ER chaperone calreticulin to the plasma membrane in both human and murine GBM cells in a concentration dependent manner. $n = 6$. **b** The membrane translocation of calreticulin induced by TMZ was transient, with peak occurring at 72-h post treatment. $n = 6$. **c** TMZ treatments did not result in significant membrane translocation of calreticulin in human or mouse normal bone marrow cells. **d** TMZ treatment promotes GBM cell phagocytosis by BM-derived phagocytes. $n = 6$. Unpaired Student's $t$ test. **e** TMZ treatment (50 μM) upregulated the mRNA expression of ER response-associated targets DDIT3, HERPUD1, and GADD45α in mouse GBM cells. $n = 4$. ** < 0.01, * < 0.05 (Unpaired Student's $t$ test). **f** Western blot showing TMZ treatment increased the expression levels of ER stress response-specific protein BiP, phospho-EIF2α, and CHOP in mouse GBM cells. TGN thapsigargin. **g** Addition of the ER stress inhibitor 4-PBT diminished the pro-phagocytosis and calreticulin translocation effect of TMZ. $n = 6$, * < 0.01 (Unpaired Student's $t$ test). Error bar = mean ± standard deviation.

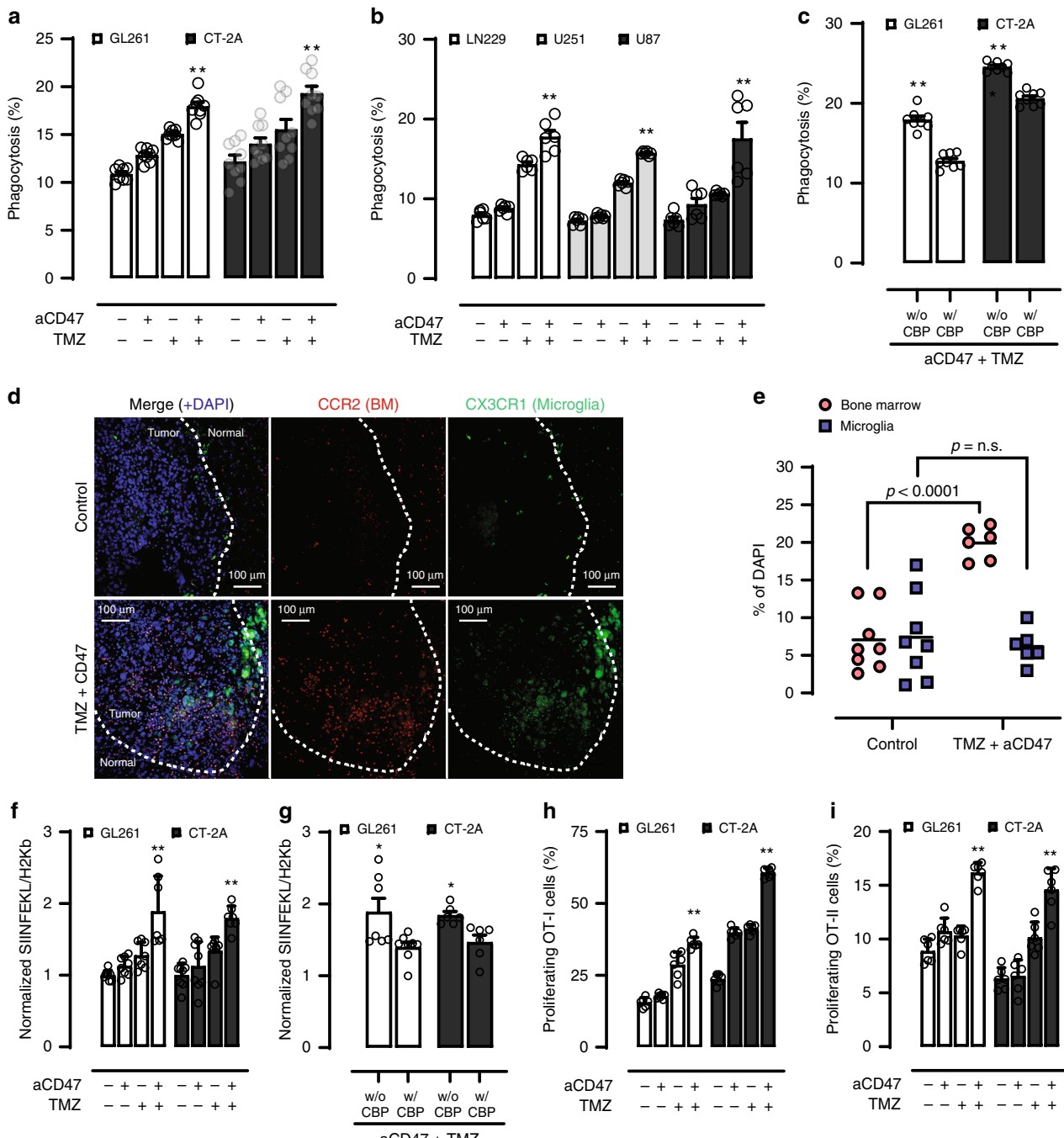

**Fig. 3 Combined TMZ and CD47 blockade enhances GBM cell phagocytosis and cross-priming of antigen-specific T cells by BM antigen presenting cells (APCs).** Combined TMZ and anti-CD47 antibody (aCD47) treatment enhances murine (**a**) and human (**b**) GBM cell phagocytosis by BM APCs. GL261, $n = 8$; CT-2A, $n = 9$; LN229, U251, U87, $n = 6$. **$p < 0.05$, one-side ANOVA with Bonferroni post hoc correction. **c** Blocking of calreticulin negated the enhanced phagocytosis effect of combined TMZ and anti-CD47 antibody treatment. $n = 8$, **$p < 0.01$, unpaired Student's $t$ test. **d** Immunofluorescence images of tumor infiltrating BM cells (red) and microglia (green). **e** Quantification of tumor-infiltrating BM cells and microglia. Control, $n = 8$; combo, $n = 6$. Unpaired Student's $t$ test. **f** Combined TMZ and anti-CD47 antibody treatment enhanced the cross-presentation of MHC-bound cOVA-derived SIINFEKL peptide on APCs. $n = 8$, **$p < 0.01$, one-side ANOVA with Bonferroni post hoc correction. **g** The enhanced antigen cross-presentation effect mediated by combined TMZ and anti-CD47 antibody was reduced with the addition of a calreticulin blocking peptide. GL261, $n = 8$, CT2A, $n = 5$. **$p < 0.01$, unpaired Student's $t$ test. **h**, **i** Combination TMZ and anti-CD47 antibody treatment enhanced cross-priming of cOVA antigen specific T cells. $n = 6$, **$p < 0.01$, one-side ANOVA with Bonferroni post hoc correction. Error bar = mean ± standard deviation.

GBM cells with BM-derived APCs and evaluated the presence of MHC-I bound cOVA-derived peptide SIINFEKL. As we expected, we noted a significant increase in the cross-presentation of cOVA-derived peptide by the APCs with combined anti-CD47 antibody and TMZ treatment (Fig. 3f). Once again, blocking of CRT significantly negated this enhanced antigen cross-presentation effect by macrophages (Fig. 3g), suggesting that membrane translocation of CRT is critical for both the observed

phagocytosis and subsequent antigen cross-presentation effect in APCs.

**Combo treatment increases T-cell priming.** Given that antigen-cross presentation by professional APC is the necessary step for T-cell priming, we next evaluated whether combination treatment promoted more efficient priming of antigen-specific T cells. To do so, we cocultured cOVA-expressing murine glioma cells treated with TMZ and anti-CD47 antibody with BM APCs. The APCs were subsequently exposed to naïve CD8+ or CD4+ T cells isolated from OT-I and OT-II transgenic mice, which express T-cell receptors that recognize cOVA-derived peptides, respectively. The combination treatment significantly increased the proliferation of OT-I and OT-II T cells by APCs, in an antigen-specific manner, as compared to either TMZ or anti-CD47 monotherapy (Fig. 3h, i), an effect that is also dependent on CRT translocation (Supplementary Fig. 11a). Examination of the proliferating T cells further revealed that a significant number of CD8+ T cells acquired surface markers that are consistent with effector or central memory phenotype (Supplementary Fig. 11b). Collectively, these findings indicate that by promoting tumor cell phagocytosis, combined TMZ and anti-CD47 treatment enhance the cross priming of antigen-specific CD8+ T cells by APCs.

**Combo therapy activates cGAS-STING pathway.** Efficient T-cell priming requires both cross-presentation of antigens and increased production of pro-inflammatory cytokines by APCs[25,26]. We found that combined TMZ and anti-CD47 treatment significantly increase the level of type I interferon mRNAs (*Ifna* and *Ifnb*) in BM APCs (Fig. 4a). Similarly, we also noted increased activating phosphorylation of the cytosolic DNA-sensor stimulator of interferon genes (STING) at serine-366, as well as phosphorylation of serines-385 and 396 residues on its downstream transcription factor interferon regulatory factor 3 (IRF3) (Fig. 4b, Supplementary Fig. 12a), which is associated with type I IFN transcription[27,28]. In addition, phosphorylation of the canonical NF-κB member p65 was also accompanied by an increased nuclear translocation of this transcription factor (Fig. 4b, c), as well increased production of pro-inflammatory cytokines in APCs (Fig. 4d)[29]. STING deficiency in APCs not only completely abrogated the phosphorylation and nuclear translocation of p65 (Fig. 4b, c, Supplementary Fig. 12b), it also significantly inhibited the NF-κB cytokine production and CD8+ T-cell priming effect (Fig. 4d, Supplementary Fig. 12c). Further, phosphorylation of p65 was minimal in tumor cells upon combination treatment, indicating that NFκB activation is restricted to APCs (Supplementary Fig. 12d). We next examined the role of cGAS-STING pathway in response to the combined TMZ and anti-CD47 treatment in vivo. The GL261 cells were injected to the brain of wild-type (WT) C57BL/6 mice and STING knockout (KO) mice and received the combination treatment. The combination treatment strongly induced CD45+ cell infiltration and IRF3 phosphorylation in the WT mice but not in the KO mice, which recapitulate our in vitro results and confirmed the lack of STING signaling in the KO mice (Fig. 4e–g). At day 20, the control treated tumors were much larger in STING KO mice than WT underscoring the importance of this pathway in tumor regulation. When treated with TMZ plus anti-CD47, a strong tumor growth inhibitory effect accompanied by increased T-cell infiltration was detected in the WT mice (Fig. 4h, i). Combination treatment also reduced tumor volume in the KO mice, however, these tumors were significantly larger as compared with WT, and were less infiltrated by CD3+ T lymphocytes (Fig. 4h, i). Together, these results suggest that STING-dependent DNA sensing pathway in APCs is essential for its T-cell priming effect in the setting of TMZ and CD47 blockade-induced tumor cell phagocytosis.

**Combo treatment activates immune responses in vivo.** To assess the antitumor effect of combined TMZ and anti-CD47 treatment on survival in vivo, we treated mice harboring syngeneic GL261 or CT-2A tumors using both a sequential and concurrent regimen (Supplementary Fig. 13a). Interestingly, the sequential arm, in which TMZ was given alone in advance followed by TMZ plus anti-CD47 antibody, resulted in significantly improved tumor growth inhibition and survival as compared to monotherapy and concurrent treatment arms in both GL261 and CT-2A models (Fig. 5a, b; Supplementary Fig. 13b, c; Supplementary Fig. 14). Tumor tissue analyses revealed a significant increase in the number of Iba1+ activated macrophages/microglia despite a minimal change in intratumoural F4/80+ cells (Fig. 5c) in the combination treatment group and was accompanied by increased serum levels of interleukin-2 (IL-2) and tumor necrosis factor alpha (TNFα) (Fig. 5d). Consistent with our previous observations, tumor tissue analyses also showed increased levels of both phospho-p65 and nuclear IRF3 (Supplementary Fig. 15), supporting STING activation in vivo.

Evaluation of adaptive immune cell profiles in the setting of combined TMZ and anti-CD47 treatment demonstrated a significant increase in both CD4+ and CD8+ T cells within the tumor (Fig. 5e). Further analyses of T-cell subsets revealed that while there was no change in T-regulatory cells (Fig. 5f), a significant increase in the number of tumor infiltrating interferon gamma (IFNγ) producing CD8+ effector T cells was noted (Fig. 5g). Depletion of CD8+ T cells using an anti-CD8 antibody nearly completely eliminated both intratumoral and systemic CD8 T cells and significantly diminished the antitumor effect of combined TMZ and anti-CD47 treatment (Fig. 5h, i). These results confirm that by stimulating innate immune responses via induced tumor cell phagocytosis by APCs, the antitumor effect of combined TMZ and anti-CD47 treatment ultimately relies on the generation of adaptive immunity mediated by cytotoxic CD8+ T cells.

The intricate relationship between innate and adaptive immune responses to produce optimal antitumor effect is further illustrated when combined TMZ and anti-CD47 treatment increased the production of IFNγ in tumors (Fig. 6a). However, when we compared rapidly progressing tumors (size > 50th percentile at 1 week after last treatment dose) with those that exhibited a more indolent course, a significant difference in the expression levels of PD-L1 and PD-1 within the extracted tumor tissue was noted (Fig. 6b). Therefore, it is possible that the innate immune activation in the setting of combination treatment produced IFNγ responses which possess antitumor effect, but also promote adaptive immune resistance via upregulation of immune checkpoints[30]. To overcome this potential resistance mechanism and improve the therapeutic efficacy of TMZ and CD47 blockade, we added an anti-PD-1 antibody in the adjuvant setting to the combined TMZ and anti-CD47 treatment arm. Animals harboring GL261 tumors, which are normally nonresponsive to PD-1 blockade, exhibited remarkable survival benefit to the triple therapy regimen, with more than 55% of animals experiencing long-term remission (Fig. 6c).

## Discussion

In this study, we show therapeutic modulation of the phagocytosis axis drastically improves the efficacy of cancer immunotherapy against poorly immunogenic tumors. Whereas previous studies utilizing antibodies to block phagocytosis checkpoints on tumor cells have yielded conflicting results[2,7], our data shows that efficient disruption of tumor cell phagocytosis and subsequent immune surveillance evasion requires the simultaneous blockade of innate phagocytosis checkpoints and

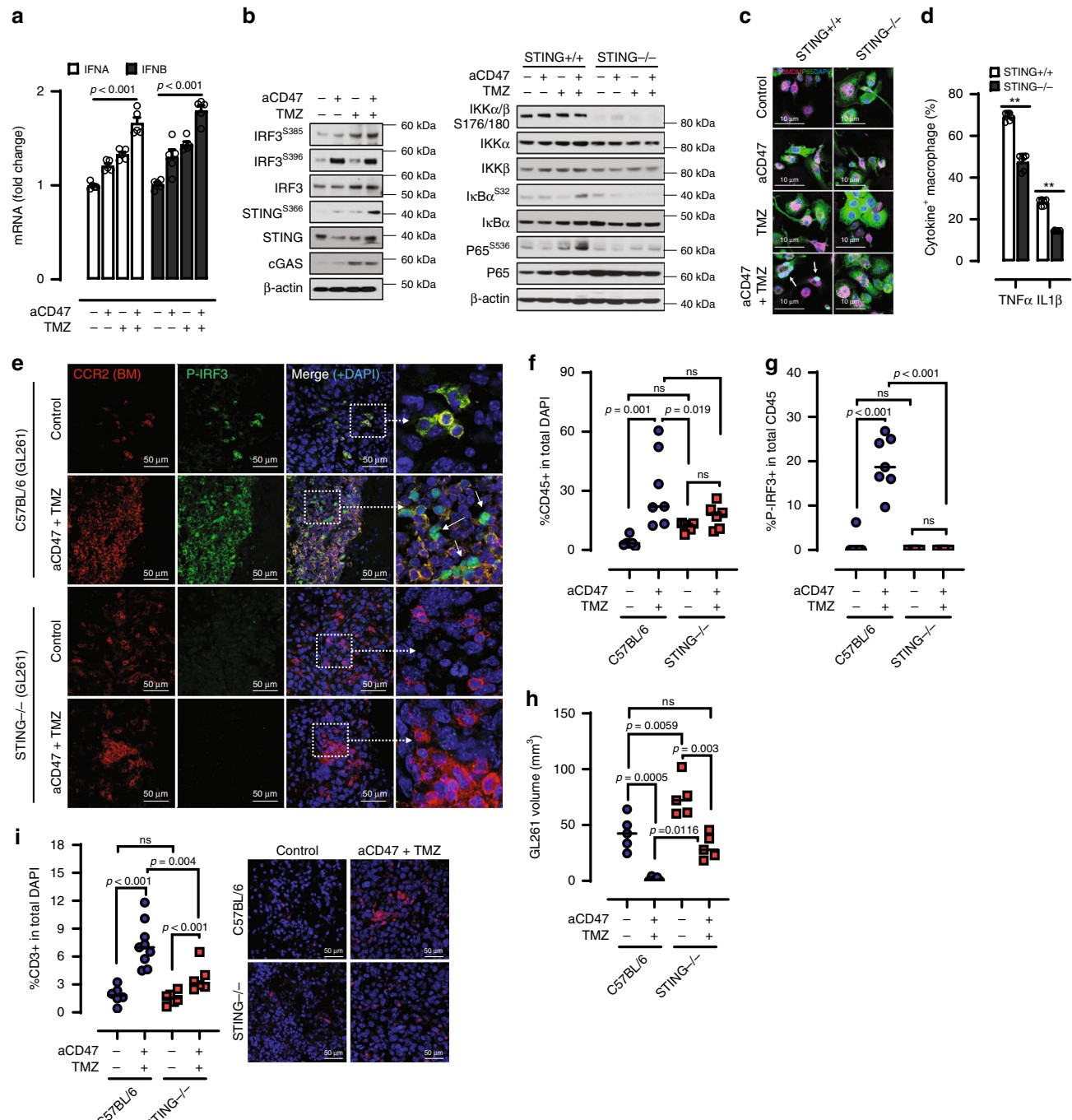

**Fig. 4 cGAS-STING pathway is essential for TMZ and anti-CD47 induced immune response. a** Combination TMZ and anti-CD47 antibody treatment against murine GBM cells (GL261) increased the production of type I interferons in APCs. IFNA interferon α, IFNB interferon β. $n = 5$, **$p < 0.01$, one-side ANOVA with Bonferroni post hoc correction. **b** Western blot showing combined TMZ and anti-CD47 antibody treatment resulted in the activation of cGAS-STING cytoplasmic DNA sensing pathways in BM APCs. **c** Combination TMZ and anti-CD47 antibody treatment promoted p65 expression and nuclear translocation in BM APCs that is dependent on STING. **d** STING activation is critical to the increased the production of NF-κB cytokine TNFα and IL1β in APCs in the setting of combined TMZ and anti-CD47 antibody treatment. $n = 6$, **$p < 0.01$, unpaired Student's $t$ test. **e** Immunofluorescence staining of GL261 tumors implanted in WT and STING KO animals. Nuclear p-IRF3 (Ser 396) in CD45+ cells is indicative of STING signaling activation. **f, g** Quantification of CD45+ cell infiltration and percentage of p-IRF3 positive CD45+, C57BL/6, $n = 7$; knockout, $n = 6$. Unpaired Student's $t$ test. **h** GL261 tumor volume in C57BL/6 or STING KO mice at day 20 following control or combination TMZ + aCD47 treatment, $n = 5$/group. **i** T-cell infiltration in GL261 tumors in C57BL/6 or STING KO mice at day 20 following combination TMZ + aCD47 treatment measured by CD3+ cells in total DAPI count per field of view (FOV), C57BL/6, $n = 9$; knockout, $n = 6$ unpaired Student's $t$ test. Representative FOV for each group on the right. All error bars = mean ± standard deviation.

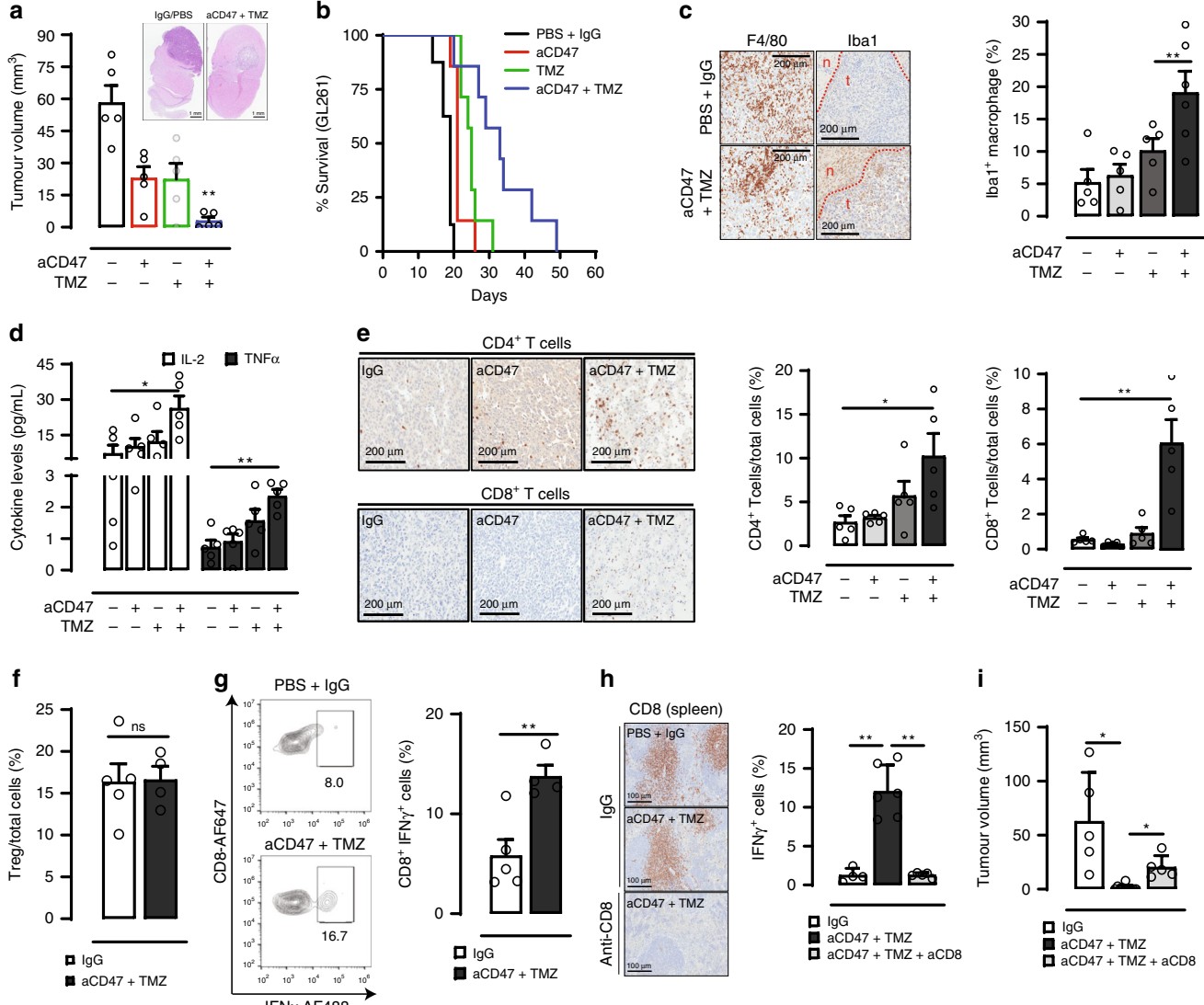

**Fig. 5 Combination of TMZ and anti-CD47 induces anti-tumor immune response in murine GBM models. a** Sequential TMZ and anti-CD47 antibody combined treatment inhibited growth of murine GBM (GL261). $n = 5$/group, **$p < 0.01$, unpaired Student's $t$ test vs. aCD47 or TMZ. **b** Sequential TMZ and anti-CD47 antibody combined treatment prolonged animal survival. Control, $n = 8$; other groups, $n = 7$. **$p < 0.05$, log-rank test. **c** Combination TMZ and anti-CD47 antibody treatment significantly increased the number of Iba1+ activated mononuclear cells within tumors, but not the total number of F4/80+ cells. Combo, $n = 6$; other groups, $n = 5$. **$p < 0.01$, unpaired Student's $t$ test. **d** Combined TMZ and anti-CD47 antibody treatment elevated the levels of peripheral blood cytokines. $n = 5$, **$p < 0.05$, one-side ANOVA with Bonferroni post hoc correction. **e** Intratumoral infiltrating CD4+ and CD8+ T cells were increased by combination TMZ and anti-CD47 antibody treatment. Scale bar = 200 μm, $n = 5$, **$p < 0.01$, one-side ANOVA with Bonferroni post hoc correction. **f, g** Combination TMZ and anti-CD47 antibody treatment did not result in significant changes to intratumoural regulatory T-cell content, but increased the number of IFNγ-producing CD8+ T cells within tumors. Control, $n = 5$; combo, $n = 4$. **$p < 0.01$, unpaired Student's $t$ test. ns not significant. **h** In vivo depletion of CD8+ T cells using an anti-CD8 antibody completely eliminated the intratumoural IFNγ+CD8+ T cells. Scale bar = 200 μm, Control, $n = 4$; combo, $n = 6$; combo + aCD8, $n = 5$. **$p < 0.01$, unpaired Student $t$ test. **i** CD8+ T cells depletion diminished the antitumor effect of combination TMZ and anti-CD47 antibody treatment. Control, $n = 5$; combo, $n = 10$; combo + aCD8, $n = 5$ *$p < 0.05$, unpaired Student $t$ test. All error bars = mean ± standard deviation.

induction of pro-phagocytosis signals. This can be achieved by conventional standard-of-care chemotherapies to activate ER stress responses that promote tumor cell phagocytosis by professional APCs and combined with CD47 blockade to generate potent T-cell priming and antitumor immune responses. Mechanistically, the linkage between innate and adaptive immune response is not only mediated by phagocytosis-induced antigen cross-presentation, but also via the activation of cytosolic DNA sensing pathways in APCs. Yet, this stimulated innate antitumor response, although desired, also carries the risk of promoting adaptive immune resistance by the upregulation of immune

checkpoints within tumors, thus further highlighting the dynamic and complex relationship between the two branches of the immune system (Fig. 6d). Therapeutically, these findings have an important implication for designing combination immunotherapies with PD-1 blockade. For example, therapeutic targeting of innate processes such as the phagocytosis axis may alter the physiology of checkpoint blockade resistant tumors to produce more favorable responses. Alternatively, checkpoint blockade may also serve as an effective adjuvant to overcome adaptive immune resistance in tumors treated with innate immune stimulating agents.

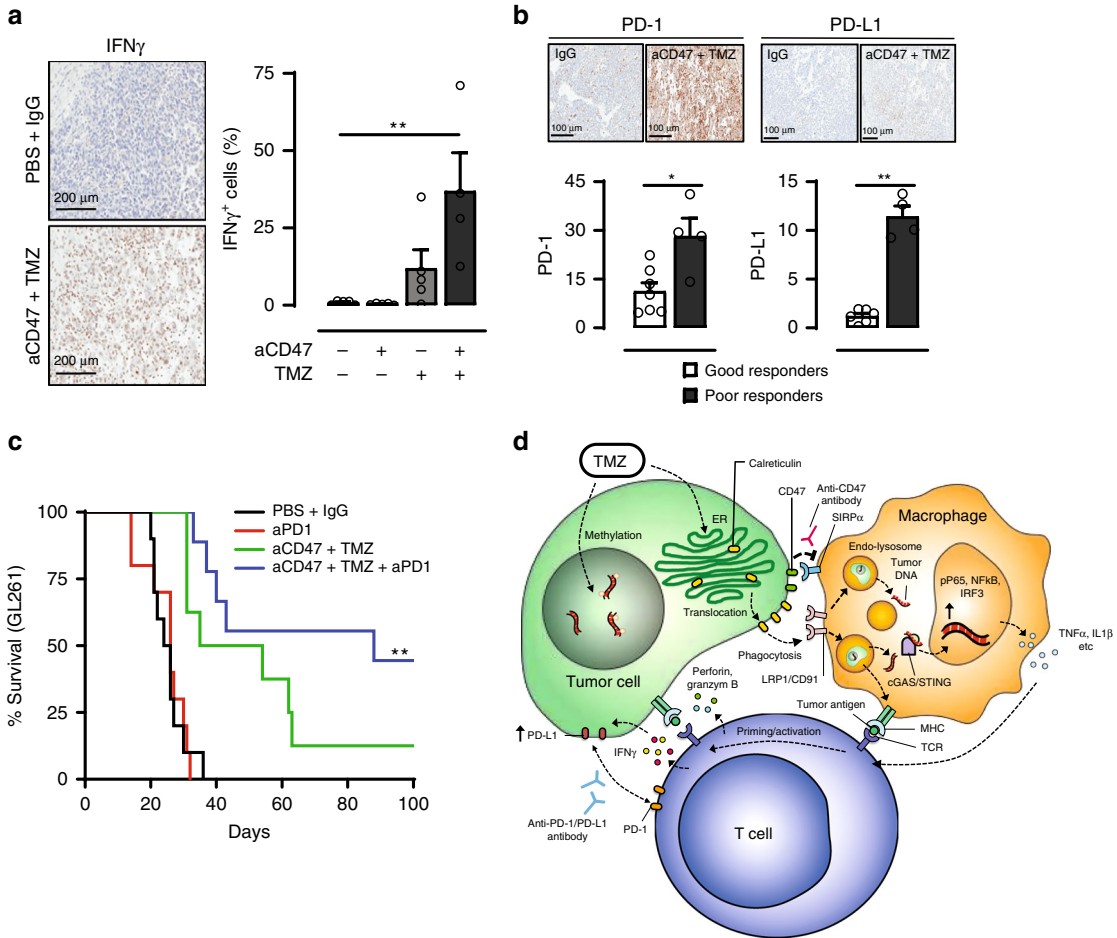

**Fig. 6 The anti-GBM effect of combined TMZ and anti-CD47 antibody treatment is augmented by adjuvant PD-1 blockade. a** Combination TMZ and anti-CD47 antibody treatment significantly elevated the level of IFNγ within tumors. Scale bar = 200 μm, Combo, $n = 4$; other groups, $n = 5$. **$p < 0.01$, one-side ANOVA with Bonferroni post hoc correction. **b** PD-1 (good responders, $n = 7$; poor responders, $n = 4$) and PD-L1 (good responders, $n = 6$; poor responders, $n = 4$) expression levels were notably elevated in GBM treated with combination TMZ and anti-CD47 antibody treatment. **$p < 0.01$, unpaired Student's t test. **c** The addition of an adjuvant anti-PD1 antibody treatment prolonged animal survival in mouse with GBM. Control and aPD-1, $n = 10$; aCD47 + TMZ, $n = 8$; aCD47 + TMZ + aPD1, $n = 9$. **$p < 0.01$, log-rank test. **d** Proposed schematic of bridging the innate and adaptive immune responses by phagocytosis induction. Calr calreticulin. TCR T-cell receptor. All error bars = mean ± standard deviation.

## Materials and methods

**Mice**. Six- to eight-week-old C57BL/6J, Tmem173[gt], CCR2[RFP]CX3CR1[GFP], OT-I CD8[+] TCR-Tg, and OT-II CD4[+] TCR-Tg mice were purchased from Jackson Laboratory, and maintained at the animal facility of Mayo Clinic in Florida, University of Florida, University of Texas Southwestern Medical Center, or MD Anderson Cancer Center in specific-pathogen-free environment. Tumor engraftment was performed at age 6–12 weeks. All animal use was approved by the Institutional Animal Care and Use Committee (Mayo Clinic in Florida, University of Florida, University of Texas Southwestern Medical Center and MD Anderson Cancer Center). All animal experiments were performed in compliance to the approved protocol and institutional policies.

**Clinical samples**. Freshly resected tumor tissue samples from patients with pathology confirmed diagnosis of primary GBM were collected under the Mayo Clinic Institutional Review Board protocol (IRB Number: 18-009866) with all identifying information removed. Informed consent were obtained from all patients. All patients were treatment naïve at the time of tissue collection.

**Cell lines**. LN229, U87, SWI-1783, G112, T98G, TP365, U251, SWI-008, U118, and D32 are human malignant glioma cell lines, and were provided by Dr. Panagiotis Anastasiadis (Mayo Clinic). All cell lines were characterized by short tandem repeat analysis. GL261 is a murine malignant glioma cell line. The mouse malignant astrocytoma cell line CT-2A was a kind gift from Dr. Thomas Seyfried (Boston College). The U138 cell line and the human monocyte cell line THP-1 was purchased from ATCC. For all cell-line authentication, none of the cell lines used in the study are listed in the Database of ICLAC. All cell lines were tested for mycoplasma contamination using a biochemical method with Hoechst staining and were found free of mycoplasma contamination.

**Reagents**. Mouse anti-CD47 blocking antibody (clone MIAP-410), human anti-CD47 blocking antibody (clone B6H12), mouse anti-PD1 antibody (clone 29F.1A12) and mouse anti-CD8 depleting antibody (clone YTS 169.4) were purchased from BioXcell (West Lebanon, NH). Temozolomide (Pharmaceutical Grade) was purchased from Cayman Chemical and prepared according to manufacturer's recommendations.

**Cell culture**. Monocytes were extracted from the BM of C57BL/6 or Tmem173[gt] mice femurs. Briefly, the back legs of euthanized mice were cut above the hip joint to access the femur. Leaving the knee and ankle joints in place, the muscle and tissue was debrided from the femur using a scalpel. Intact femurs were resected, transferred to a laminar flow hood, externally sterilized with 70% ethanol for 10 s, and transferred to cold sterile phosphate-buffered saline (PBS). Both ends of the femur were cut as close to the joint end as possible using sterile scissors, and BM was flushed out with ice cold PBS using a syringe. Cells were filtered through a 70 μm cell strainer, and maintained in DMEM medium supplemented with 30% L929 (Stony Brook Cell Culture/Hybridoma Facility), 20% fetal bovine serum (FBS), 1% sodium pyruvate, and 1% penicillin/streptomycin (P/S). THP-1 cells were cultured in RPMI 1640 (ThermoFisher) containing 10% FBS and 1% P/S. All murine and human glioma cell lines were cultured in DMEM (ThermoFisher) supplemented with 10% FBS and 1% P/S. Cell cultures were incubated at 37°C in humidified conditions equilibrated with 5% $CO_2$.

**RNA extraction and quantitative real-time RT-PCR**. Total RNA content from $1 \times 10^5$ cells were extracted using PureLink® RNA Mini Kit (Applied Biosystems) per manufacturer's protocol and cDNA conversion was done using High-Capacity cDNA RT Kit (ThermoFisher). Quantitative PCR reactions and analysis were performed using ViiA 7 (ThermoFisher). For BM APC experiments, cells were

enriched using CD45 magnetic beads. TaqMan®FAM™ dye-labeled probes including *GAPDH* (Hs02786624_g1, Mm99999915_g1), *GADD45A* (Hs00169255_m1, Mm00432802_m1), *DDIT3* (Hs00358796_g1, Mm01135937_g1), *HERPUD1* (Hs01124269_m1, Mm00445600_m1), *IFNA1* (Mm03030145_gH), and *IFNB1* (Mm00439552_s1) (ThermoFisher). GAPDH was used to normalize signal expression. Fold change comparison were performed between control and treated samples using the ΔΔCT method as described[31].

**Western blotting.** Cells were lysed with radioimmunoprecipitation assay buffer (ThermoFisher) containing protease inhibitor cocktail (Cell Signaling), and Halt™ Phosphatase inhibitor (ThermoFisher). After gel electrophoresis (NuPAGE Bis-Tris Gel, ThermoFisher), proteins were transferred onto Immobilon-PSQ PVDF membranes (ThermoFisher), and blocked in membrane blocking solution (Invitrogen) for 1 h at RT. Blot was incubated with primary antibody overnight at 4 °C. The primary antibody was diluted in buffer according to manufacturer's specification. The primary antibody was detected by the appropriate horseradish peroxidase-conjugated secondary antibody (Jackson ImmunoResearch) followed by incubation with Pierce ECL Western Blotting Substrate (Thermo Scientific). Antibodies include: BiP (Clone C50B12; Cell Signaling, #3177, 1:1000), pEIF2a$^{ser51}$ (Cell Signaling, #3398, 1:500), EIF2a (Cell Signaling, #5324, 1:1000), CHOP (Clone L63F7; Cell Signaling, #2895, 1:1000), β-actin (Clone 13E5; Cell Signaling, #4970, 1:2000), pIRF3$^{S385}$ (pAb; ThermoFisher, #PA5-36775, 1:500), pIRF3$^{S396}$ (Clone D6O1M; Cell Signaling, #29047, 1:500), IRF3 (pAb; ThermoFisher, #PA5-87506, 1:500), pSTING$^{S366}$ (pAb; ThermoFisher, #PA5-105674, 1:500), STING (Clone D2P2F; Cell Signaling, #13647, 1:500), cGAS (Clone D3O8O; Cell Signaling, #31659, 1:500), MGMT (pAb; abcam, #ab108630, 1:500), IKKα (Clone 3G12; Cell Signaling, #11930, 1:1000), IKKβ (Clone D30C6; Cell Signaling, #8943, 1:1000), p-IKKα/β$^{ser176/180}$ (Clone 16A6; Cell Signaling, #2697, 1:1000), p-P65$^{ser536}$ (Clone 93H1; Cell Signaling, #3033, 1:1000), IκBα (Clone L35A5; Cell Signaling, #4814, 1:1000), p-IκBα$^{ser32}$ (Clone 14D4; Cell Signaling, #2859, 1:1000), P65 (Clone D14E12; Cell Signaling, #8242, 1:2000). All original uncropped film images are provided in Supplementary Fig. 17.

**Gene expression.** pLenti-cOVA construct was a gift from Dr. Charles K. Chan at Stanford University. Lentivirus was prepared in 293FT progenitor cells (Invitrogen) using Lipofectamine®3000 and ViraPower packaging mix (ThermoFisher). For lentiviral transduction, tumor cells were plated at 1 × 10⁶ cells in 60 mm culture dish. Media was replaced with 2.5 mL complete media containing 10 μg per mL Polybrene (EMD Millipore). 1 mL of lentivirus was added in a drop-wise fashion to cells. Cells were incubated overnight. Cells were washed 1× with PBS and complete media was added. cOva-expressing cells were used for experimentation 48-h post infection. MGMT retroviral expression vector was purchased from DNASU. Virus was prepared in Phoenix-AMPHO (ATCC) cells using Lipofectamine®3000 (ThermoFisher). Retroviral transduction of tumor cells was performed as described for lentiviral transduction. After 48 h, puromycin (Sigma) selection was performed, followed by clonal selection. Protein expression validation was performed using either IF or western blot.

**THP-1 differentiation.** THP-1 cells were cultured in RPMI media containing 10% FBS and 2 mmol/L L-glutamine. 200 nM phorbol 12-myristate 13-acetate (PMA, Sigma) was added to cells for 3 days. After 3 days, regular RPMI (no PMA) was added for an additional 2 days. Differentiated THP-1 cells were used for human phagocytosis assays.

**Phagocytosis assay.** Carboxyfluorescein succinimidyl ester (CFSE)-labeled (CellTrace™, ThermoFisher)-labeled tumor cells were plated at 5 × 10⁵ cells/well and allowed to adhere for 2 h in 12-well culture treated plates (Costar). Cells were treated with TMZ for 72 h or with cisplatin for 48 h, after which MIAP-410 was added for an additional hour. 1.5 × 10⁶ eFluor®670 (eBioscience) phagocytes (murine: BM-derived APCs, human: differentiated THP-1) were cocultured with treated tumor cells for 4 h at 37 °C. Cells were collected on ice, washed 2× with cold PBS, and suspended in HBSS containing 1% bovine serum albumin (BSA). Phagocytosis was analyzed using a CytoFLEX Flow Cytometer (Beckman Coulter) and calculated as the percentage of CFSE⁺ cells within total eFluor®670⁺ phagocyte population. For calreticulin neutralizing assay, calreticulin blocking antibody (ThermoFisher) was added at 10 μg per ml one hour prior to coculture. For immunofluorescence, 1 × 10⁴ eFluor670-labeled tumor cells were suspended with 3 × 10⁴ CFSE-labeled phagocytes in 200 μL phenol-free matrigel (Corning) plated in 12-well optical viewing chamber plates (MatTek). After polymerization of the gels, 3-D cultures were equilibrated in complete phenol-free DMEM (ThermoFisher) overnight. TMZ was added for 72 h followed by an additional 24 h MIAP-410 treatment. Images were taken using a Zeiss LSM 710 laser-scanning microscope with a 20x objective, NA 1.2 (Carl Zeiss).

**Microglia phagocytosis.** Primary microglia were isolated from P3–P5 pups. In a laminar flow hood, pups were placed on ice until unresponsive (~5 min), quickly sterilized in 70% ethanol, and decapitated. Brain tissue was resected, meninges and cerebellum removed, and transferred to a 50 mL conical tube containing ice-cold PBS. Tissue was manually dissociated using a 1000 μL pipettor, and filtered through

a 100 μm cell strainer. Cells were cultured in poly-D-Lysine coated culture dishes (Corning) with DMEM-high glucose (ThermoFisher) containing 10% FBS, 1% P/S, and 25 ng per mL recombinant mouse granulocyte-macrophage colony-stimulating factor (R&D Systems). Media was changed once at day 5. After 9 days, microglia cells were collected using Cell Dissociation Solution (Sigma) for phagocytosis assays. GL261 cells were stained with CFSE proliferation dye and treated with MIAP-410 and/or TMZ. Primary microglia cells were added to treated GL261 cells at 3 × 10⁵ per well and cocultured for 3 h at 37 °C. Cells were collected on ice, washed 2× with cold PBS, and suspended in HBSS containing 1% BSA. Cells were stained with anti-mouse CX3CR1-AF700 (clone SA011F11, BioLegend, #149036, 0.25 μg/10⁶ cells), P2X7R-PC7 (clone 1F11, BioLegend, #148708, 0.8 μg/10⁶ cells), and Sytox™ Blue dead cell stain (ThermoFisher). Microglia phagocytosis was determined as CFSE-positive cells within total microglia fraction identified as P2X7R⁺CX3CR1⁺ dual positive cells.

**Antigen presentation assay.** GBM-cOva cells were plated and treated as described in phagocytosis assay, and cocultured with CFSE-labeled BM-derived APCs for 24 h. Cells were stained on ice for 1 h with APC-labeled anti-SIINFEK6/H-2Kb (eBioscience, #17-5743-82, 0.25 μg/10⁶ cells) which detects ovalbumin-derived peptide SIINFEKL bound to H-2Kb of MHC class I, but not with unbound H-2Kb or H-2Kb bound with an irrelevant peptide. Antigen presentation was assessed using a CytoFLEX Flow Cytometer (Beckman Coulter) and was measured as the percentage of APC⁺ cells within CFSE⁺ phagocytes. For immunofluorescence, tumor and CFSE-phagocytes were plated on chamber slides (LabTEK) and treated as described. Cells were fixed with 4% w/v formaldehyde for 20 min at RT, washed 3× with PBS, and incubated with APC-labeled anti-SIINFEKL/H2Kb antibody (eBioscience) or P65 (Cell Signaling) for 60 min at RT in HBSS containing 1% BSA. Cells were washed 3× with PBS, and slides were coverslipped using ProLong Gold mounting reagent (Life Technologies). For NFκB studies, DAPI (Thermo Fisher) nuclear stain was added during the final wash before the slides were coverslipped. Images were taken using a Zeiss LSM 710 laser-scanning microscope with a 40× oil emersion objective, NA 1.2 (Carl Zeiss).

**T-cell activation.** GBM-cOVA cells were plated and treated as described in phagocytosis assay, and cocultured with unlabeled BM-derived APCs for an additional 24 h. Naïve T cells were isolated from OT-I (CD8) or OT-II (CD4) murine spleen (Jackson Laboratory), labeled with eFluor670 proliferative dye (eBioscience), and added to cocultures for an additional 72 h. For proliferation analysis, cells were collected and stained with CD3, CD4, and CD8 (BioLegend). T-cell proliferation was determined using a CytoFLEX Flow Cytometer (Beckman Coulter), where the T cell population demonstrating dilution of eFluor670 proliferative dye over total CD4 (CD3⁺CD4⁺eFluor670⁺) or CD8 (CD3⁺CD8⁺eFluor670⁺) populations was calculated. For T-cell maturation, cell isolate was additionally stained using CD44 (clone IM7; BioLegend) and CD62L antibodies (clone MEL-14; BioLegend). Maturation was assessed by the ratio of CD44 to CD62L expression, where naïve T cells are CD44$^{lo}$CD62L$^{hi}$, effector T cells are CD44$^{hi}$CD62L$^{lo}$, and memory T cells are CD44$^{hi}$CD62L$^{hi}$.

**T-cell isolation.** Spleens from OT-I or OT-II transgenic mice were carefully removed and washed twice with sterile PBS on ice under sterile conditions. Tissue was dissociated through a 70 μM cell strainer using mechanical force. ACK lysis (2 mL per spleen) (ThermoFisher) was performed to remove erythrocytes. T-cell isolation was performed using EasySep™ Mouse T cell isolation kit (STEMCELL Technologies) according to manufacturer protocol.

**Cytokine assay.** Serum was isolated from whole blood collected via heart puncture from C57BL/6 mice at the study endpoint. Total serum was diluted 1:2 in PBS prior to ELISA determination of concentrations of IL-2 and TNFα (BioLegend) via manufacturer's instructions.

**Tumor growth and treatments.** Buprenorphine (0.1 mg per kg) was given as analgesic immediately prior to surgery and 24 h post op. Animals were anesthetized using isofluorane. Briefly, the scalp was swabbed with povidone-iodide, and a midline incision was made. A burr hole was drilled through the skull at the following coordinates: 2 mm posterior to the Bregma, and 2 mm lateral from the midline. Stereotaxic injection of GL261 or CT-2A cells (1 × 10⁵ in 2 μl total volume/mouse) into the caudate putamen was done using a Hamilton 5 μl syringe at a depth of 3 mm. After injection, the incision was sutured (Doccol Corporation). TMZ/MIAP-410 regimen was administered as follows: metronomic TMZ (20 mg per kg) was injected intraperitoneally days 7–9. TMZ (80 mg per kg) and MIAP-410 (100 μg) were injected intraperitoneally days 11, 13, and 15. PD-1 antibody (100 μg) was administered intraperitoneally on days 16, 18, and 20. For tumor volume, IHC, and immunological analysis, brain tissue was harvested on day 20 or 22. Kaplan–Meier survival comparison was performed using log-rank (Mantel-Cox) test (Graphpad Prism software version 7.03)

**CD8⁺ T-cell depletion assay.** C57/BL6 mice were injected with 300 μg of anti-CD8 mAbs (In VivoPlus® clone YTS 169.4; BioXCell) intraperitoneally every 72 h

beginning four days prior to sequential combination treatment and continued every 72 h for the duration of the study. Depletion was confirmed by IHC analysis of murine spleen isolated upon study completion.

**Tumor dissociation.** Tumor tissue was resected from fresh murine brains and dissociated using Mouse Tumor Dissociation Kit in autoMACS® Pro Separator (Miltenyi) per manufacturer protocol. Leukocyte populations were isolated using CD45 MicroBeads (Miltenyi) per manufacturer protocol, performed in LS Columns (Miltenyi) on QuadroMACS magnet (Miltenyi). Sample fixation and staining was performed using PerFix-nc kit (Beckman Coulter).

**Flow cytometry.** Flow cytometry analysis was performed on a CytoFLEX flow cytometer (Beckman Coulter). Fluorescently labeled antibodies, include CD47 (clone B6H12.2; abcam, #ab134484, 5ul/10^6 cells & clone MIAP310; BioLegend, #127507, 0.5 µg/10^6 cells), CD45 (clone 30-F11, BioLegend, #103121, 0.25 µg/10^6 cells), CD3 (clone 17A2; BioLegend, #100201, 0.25 µg/10^6 cells), CD4 (GK1.5; BioLegend, #100425, 0.25 µg/10^6 cells), CD8 (53-6.7; BioLegend, #100727, 0.25 µg/ 10^6 cells), FoxP3 (clone MF-14; BioLegend, #126401, 0.25 µg/10^6 cells), CD25 (clone 3C7; BioLegend, #101903, 0.1 µg/10^6 cells), IFNγ (clone XMG1.2; BioLegend, #505815, 0.25 µg/10^6 cells), CD11b (clone M1/70; BioLegend, #101219, 0.25 µg/10^6 cells), CD11c (clone N418; BioLegend, #117301, 0.5 µg/10^6 cells), MHC II (clone M5/114.15.2; BioLegend, #107615, 0.25 µg/10^6 cells), F4/80 (clone BM8; BioLegend, #123119, 0.25 µg/10^6 cells), Gr1 (Clone RB6-8C5; BioLegend, # 108419, 0.25 µg/10^6 cells), and calreticulin (pAb; US Biological, #033154, 1 µL/10^6 cells). SYTOX® Blue or propidium iodide stain (ThermoFisher) was used to discriminate live from dead cells. Specific populations were identified as follows: CD4 T cells (CD45+CD3+CD4+), CD8 T cells (CD45+CD3+CD8+), $T_{reg}$ (CD45+CD3+CD4/8+CD25+FoxP3+), $T_{eff}$ (CD45+CD3+CD4/8+IFNγ+), DCs (CD45+CD11c^hiMHCII^hi), Mac (CD45+CD11c^neg-modMHCII^neg-modF4/80+), MDSC (CD45+CD11b+CD11c^neg-modMHCII^neg-modF4Gr1+). All gating strategy are shown within the figures or summarized in Supplementary Fig. 18.

**Apoptosis assay.** The glioma cells were plated in six-well plates and incubated overnight to adhere. Then the media was changed to complete culture media with temozolomide. The cells were treated for 72 h then collected by trypsinization and stained by Annexin V-FITC staining kit (Abcam). The cells were analyzed by flow cytometry and the FITC+ cells were determined as apoptotic. Representative gating strategy to determine apoptotic cells is shown in Supplementary Fig. 7d.

**Cell proliferation assay.** The glioma cells were plated in 96-well plates. Fresh complete media with temozolomide were changed on the next day. The cell numbers were determined by the CellTiter 96 AQueous Non-Radioactive Cell Proliferation Assay kit (Promega) after 24, 48, and 72 h of treatment. The OD490 values were normalized to the value at 0 h baseline for each cell line respectively.

**Immunohistochemistry.** Buffered formalin (Fisher) fixed murine brain and human brain samples were mounted on slides, blocked with Diluent (Dako), and stained using the following primary antibodies: CD47 (clone B6H12.2; thermofisher, #14-0479-82, 1:50), F4/80 (clone SP115; abcam, #ab111101, 1:500), Iba1 (pAb; abcam, #ab5076, 1:2000), CD4 (clone 4SM95; eBioscience, #14-9766-82, 1:100), CD8 (clone 4SM15; eBioscience, #14-0808-82, 1:50), IFNγ (pAb; Bioss, #BS-0480R, 1:50), PD-L1 (clone D5V3B; Cell Signaling, #64988, 1:100), PD-1 (clone D7D5W; Cell Signaling, #84651, 1:800), and IRF3 (pAb; ThermoFisher, #PA5-87506, 1:100). Analysis was performed using Imagescope software (Aperio). Where indicated results are presented as % positive cells within total population (identified by total nuclear count); or H-score, where staining intensity level is calculated using the formula $[1 \times (\% \text{ cells } 1+) + 2 \times (\% \text{ cells } 2+) + (\% \text{ cells } 3 +)]$, in the region of interest. For immunofluorescent staining, fresh frozen tumor tissue was embedded in OCT compound (Fisher). 10 µM sections were cut using a cryostat (Microm), fixed in buffered formalin (Fisher) for 10 min, blocked with Diluent (Dako), and stained overnight at 4 C with the following antibodies: CD45 (clone 30-F11, ThermoFisher, #14-0451-82, 1:100), CD3 (clone 145-2C11, ThermoFisher, #14-0031-82, 1:50), phospho-IRF3 S396 (clone D6O1M, Cell Signaling, #29047, 1:50). Slides were washed 3× in PBS containing 0.1% Tween-20 (Sigma) followed by fluorescent secondary antibody (ThermoFisher) labeling with DAPI stain for 2 h at RT. Slides were imaged using a Nikon A1RMP Confocal Microscope, and analyzed using Nikon NIS-Elements v4.51 software.

**Statistical analysis.** Where indicated, statistical analysis was performed using one- or two-way ANOVA variance, or unpaired two-tailed Student's $t$ test (Graphpad Prism software version 7.03). Asterisks are used to indicated statistical significance (where *$p < 0.05$, **$p < 0.01$, ***$p < 0.001$, ****$p < 0.0001$). n.s. is nonsignificant, where $p > 0.05$. Error bar = mean ± standard deviation in all graphs unless otherwise specified.

**Reporting summary.** Further information on research design is available in the Nature Research Reporting Summary linked to this article.

## Data availability

All data is presented in the Figures and Supplementary Figs. The source data that was used to generate the figures is provided as a Source Data file.

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

## Acknowledgements
This work was supported by grants from the National Institute of Neurological Disorders and Stroke Grant R01 NS104315 (B.Y.S.K.), the Cancer Prevention and Research Institute of Texas RR180017 (W.J.), the American Brain Tumor Association DG1900021 (W.J.), the National Cancer Institute K08 CA241070 (W.J.), and the Preston A. Wells, Jr. Endowment at the University of Florida (D.A.M.). The authors thank Joshua Knight and Laura Lewis-Tuffin and Yuka Gataki for their technical assistance, as well as Christine Wogan from the University of Texas MD Anderson Cancer Center for editorial assistance.

## Author contributions
W.J. and B.Y.S.K conceived the study and supervised the research. C.A.V., Y.W., Y.Q., H.Y., H.Z., X.L., and M.Y. performed the majority of the experiments and generated the data. C.A.V., Y.W., W.J., and B.Y.S.K. prepared the figures. C.A.V., Y.W., Y.Q., H.Y., H.Z., X.L., M.Y., Z.Y., W.D., K.B., C.K.C., A.S.L., S.R., K.Y., A.J., and D.A.M. analyzed the data and interpreted the results. C.A.V., Y.W., W.J., and B.Y.S.K. wrote the paper.

## Competing interests
The authors declare no competing interests.
