## [Peer Review File · Nature Communications]

Reviewers' comments:

Reviewer #1 (Remarks to the Author): Expert in GBM and immunotherapy

In their manuscript entitled, "Therapeutic modulation of glioblastoma phagocytosis axis bridges innate and adaptive antitumor immunity," von Roemeling et al highlight that combining CD47 blockade with temozolomide may provide some mechanistic synergy that bridges innate and adaptive immunity to improve anti-tumor immune responses in murine glioblastoma models. The experiments conducted are detailed and thorough and the mechanistic findings intriguing. Statistical analysis is valid. Most concerns surround the novelty of the work, the appropriateness of the models, and the applicability beyond glioblastoma. Some of these concerns and polite suggestions for improving the impact of the work are detailed below:

- 1) As the authors themselves highlight, targeting CD47 is not a novel therapeutic approach in cancer. Thus, the incremental advance here is the combination with temozolomide, which is the standard chemotherapeutic for glioblastoma. The authors note that the actions of temozolomide may form a rationale for additive effects with CD47 blockade, and work to demonstrate those additive mechanistic effects. It remains unclear, however, whether temozolomide is the best agent to demonstrate these additive benefits, as no other chemotherapeutics were tested. It is similarly unclear, then, whether glioblastoma is the best tumor with which to observe this combination. The impact of this work would be made larger by demonstrating an anti-tumor "principle" that extends beyond glioblastoma and temozolomide. Otherwise, this paper is simply a demonstration that combining two previously existing strategies may produce additive effects in glioblastoma. While interesting and well-demonstrated, the novelty of such a finding at this time remains limited.
- 2) Along similar lines, it is unclear whether the findings advanced by the authors are well-suited to the intracranial microenvironment or to peripheral tumor environments as well. There are of course peculiarities to CNS immunity, and those peculiarities are only mildly explored in this manuscript (Supp Fig 9 investigates the in vitro effects on microglia). This lack of exploration unfortunately highlights a weakness in the chosen in vitro systems: nearly all the experiments in Fig 3 are conducted with bone marrow-derived APC, a cell type with questionable relevance to the intracranial glioblastoma environment. If the goal is to demonstrate applicability to glioblastoma, then the authors should conduct the experiments in Fig 3 in vivo, administering the drugs, harvesting tumor, and determining which APC in the microenvironment is impacted (i.e. macrophages vs microglia, etc). This they did to some extent in Fig 4, but the details of what is actually occurring in the brain remain unclear, as they did not address well the APC populations – an important missing in vivo finding given the subject of the paper. Fig 3 does little to reveal actual intracranial mechanistic consideration in its current form as it is in vitro and utilizing a potentially non-relevant cell population.
- 3) In vivo experiments should not just be conducted with GL261 as this is a highly criticized murine glioma model, with questionable immunologic relevance. Both mechanistic and survival experiments should be done side by side in the CT2A model, to permit comparisons of the data across multiple models.
- 4) As a smaller concern, it is not clear that improving a phagocytosis index from 10% to 20% is altogether relevant. Although the authors do demonstrate improved antigen presentation (although again unfortunately only in BM APC), some demonstration that such an improvement is truly important might be in order. It is acknowledged however that that is likely a difficult task.

Reviewer #2 (Remarks to the Author): Expert in phagocytosis

This is an important study regarding a combination therapy of CD47 blockade and temozolomide. The authors revealed a mechanism of TMZ-induced CRT exposure on glioma cells which induced phagocytosis by macrophages and activation of APC functions of macrophages, and proposed a novel treatment approach for targeting glioma. The research topic is very important and the

experiments were very well designed and performed.

Despite the enthusiasm, a few concerns need to be addressed by the authors, before the manuscript can be considered for a publication in Nature Communications.

1. Temozolomide is one of the standard malignant gliomas treatment chemotherapy drugs. TMZ treatment elicits apoptosis and autophagy of glioma cells as documented in previous studies. While an experiment was performed in the study (Fig.S6) to show TMZ didn't significantly induce apoptosis, additional experiments are needed to address 1) what are the differences regarding the doses and timing of TMZ treatment between this study and previous studies when TMZ induced apoptosis in glioma cells; 2) what would be the threshold when TMZ induces apoptosis in the glioma cells used in the manuscript; 3) What would be the effects on cancer cell phagocytosis and APC function of macrophages when TMZ reaches the concentration to induce significant apoptosis; 4) the authors should comment how the concentrations of TMZ used in in vitro experiments were translated to the doses used in the in vivo treatment, and how the doses of TMZ used in this study are relevant to its clinical use;
2. Antitumor activity of TMZ is dependent on MGMT expression. The authors tested the effects of MGMT overexpression in a cell line that does not endogenously express MGMT. It would be more convincing if the authors could examine MGMT expression in multiple lines used in the study and investigate if there is a correlation between endogenous MGMT expression, resistance to TMZ-induced CRT translocation and susceptibility to phagocytosis;
3. The authors should comment on why TMZ treatment doesn't induce CRT translocation in normal brain tissue cells;
4. Does TMZ treatment have a direct effect on cell proliferation of the glioma cells?

My colleagues and I would like to thank you for considering our recent work for publication in Nature Communications. We also would like to thank the reviewers for spending the time and energy to review our submitted manuscript. We greatly appreciate the constructive feedback from the reviewers to improve our work. We have made every attempt to answer and address point-by-point the issues raised by the reviewers. In the following letter, we will address the reviewers' comments and feedback one-by-one. Our responses are typed in blue.

Reviewer #1 (Remarks to the Author): Expert in GBM and immunotherapy

In their manuscript entitled, "Therapeutic modulation of glioblastoma phagocytosis axis bridges innate and adaptive antitumour immunity," von Roemeling et al highlight that combining CD47 blockade with temozolomide may provide some mechanistic synergy that bridges innate and adaptive immunity to improve anti-tumor immune responses in murine glioblastoma models. The experiments conducted are detailed and thorough and the mechanistic findings intriguing. Statistical analysis is valid. Most concerns surround the novelty of the work, the appropriateness of the models, and the applicability beyond glioblastoma. Some of these concerns and polite suggestions for improving the impact of the work are detailed below:

1) As the authors themselves highlight, targeting CD47 is not a novel therapeutic approach in cancer. Thus, the incremental advance here is the combination with temozolomide, which is the standard chemotherapeutic for glioblastoma. The authors note that the actions of temozolomide may form a rationale for additive effects with CD47 blockade, and work to demonstrate those additive mechanistic effects. It remains unclear, however, whether temozolomide is the best agent to demonstrate these additive benefits, as no other chemotherapeutics were tested. It is similarly unclear, then, whether glioblastoma is the best tumor with which to observe this combination. The impact of this work would be made larger by demonstrating an anti-tumor "principle" that extends beyond glioblastoma and temozolomide. Otherwise, this paper is simply a demonstration that combining two previously existing strategies may produce additive effects in glioblastoma. While interesting and well-demonstrated, the novelty of such a finding at this time remains limited.

We thank the reviewer for the excellent suggestion. In summary, our proposed mechanism of synergy ensuing temozolomide treatment of glioblastoma cells derives from induced endoplasmic reticulum stress caused by DNA damage in direct response to the alkylating effects of the drug. This promotes the translocation of calreticulin protein to the plasma membrane where it engages the LRP1 receptor to enhance phagocytic uptake. We also showed that the effect of TMZ highly depends on the MGMT deficiency (**Supplemental Fig. 8**), which is very common among glioma. We acknowledge that this phenomenon may not be exclusive to temozolomide, nor exclusive to glioblastoma cancer cells.

To test this theory, we examined if other chemotherapeutic agents commonly used in cancer treatment and has known ER stress-inducing property such as cisplatin (which also causes DNA damage) can similarly induce calreticulin translocation to the plasma membrane and promote tumour cell phagocytosis (*Oncogenesis* (2017) 6, e373). We found that cisplatin treatment for 48 hours induced ER-stress of the GL261 cell line and calreticulin translocation (**Supplemental Fig. 9a**). Cisplatin also increased phagocytosis of the GL261 cells, the phagocytosis was further enhanced by anti-CD47 treatment. Since cisplatin is an approved agent for many other solid tumors such as head and neck and breast cancers, we next tested our hypothesis in murine breast cancer cell E0771, which was derived from a spontaneously developing medullary breast adenocarcinoma (C57BL/6). The E0771 cells demonstrated high level of calreticulin translocation after 48 hours of cisplatin treatment. We also confirmed by western blot that the ER-stress markers, including p-eIF2 α , BiP and CHOP, were induced by cisplatin (**Supplemental Fig. 9c-e**). Cisplatin treatment itself significantly increased E0771 phagocytosis by bone-marrow derived macrophages, which was slightly further enhanced by anti-CD47 (**Supplemental Fig. 9f-g**). Despite the fact that the combo treatment worked better than each treatment used alone for both GL261 and E0771 models, the combination effects of cisplatin plus anti-CD47 were less robust compared with TMZ plus anti-CD47 (**Fig. 3a,b**). The difference might be the result of the different potency and mode of action of DNA-damage caused by the two agents, or potentially due to the susceptibility to anti-CD47 induced phagocytosis among different cancer models. Ultimately, the “best” chemotherapy to induce prophagocytic effect of tumour cells in concert with CD47 blockade will need to be tailored to the specific cancer models used.

2) Along similar lines, it is unclear whether the findings advanced by the authors are well-suited to the intracranial microenvironment or to peripheral tumor environments as well. There are of course peculiarities to CNS immunity, and those peculiarities are only mildly explored in this manuscript (Supp Fig 9 investigates the in vitro effects on microglia). This lack of exploration unfortunately highlights a weakness in the chosen in vitro systems: nearly all the experiments in Fig 3 are conducted with bone marrow-derived APC, a cell type with questionable relevance to the intracranial glioblastoma environment. If the goal is to demonstrate applicability to glioblastoma, then the authors should conduct the experiments in Fig 3 in vivo, administering the drugs, harvesting tumor, and determining which APC in the microenvironment is impacted (i.e. macrophages vs microglia, etc). This they did to some extent in Fig 4, but the details of what is actually occurring in the brain remain unclear, as they did not address well the APC populations – an important missing in vivo finding given the subject of the paper. Fig 3 does little to reveal actual intracranial mechanistic consideration in its current form as it is in vitro and utilizing a potentially non-relevant cell population.

Thank you for your very important suggestion. To better answer the question of which innate immune cell is responsible for governing therapeutic response to glioblastoma, we have repeated the combination temozolomide and anti-CD47 antibody treatment in GL261 tumors implanted in the CCR2^{RFP}CX3CR1^{GFP} (B6.129(Cg)-Cx3cr1^{tm1Litt}Ccr2^{tm2.1ffc}/JernJ) mouse model. In this model, CX3CR1 is labeled with green

fluorescent protein (GFP), and CCR2 is labeled with red fluorescent protein (RFP). Resident microglia demonstrate high expression of CX3CR1, but lack CCR2. Therefore, they are predominantly GFP positive. Bone marrow derived monocytes express high levels of CCR2 and low expression of CX3CR1, and therefore appear as RFP single or RFP/GFP dual positive. This has allowed us to clearly stratify which immune cell population is present within and around the tumor. Our data clearly show increased tumor infiltration of GL261 tumors by bone marrow derived monocytes following combination treatment, and little change in both quantity and localization of microglia (**Fig. 3d,e**). This new information strongly suggests that bone marrow derived cells constitute the primary APC population to respond to the TMZ and anti-CD47 combination treatment. Therefore we believe that the data presented in the manuscript utilizing BMDM are indeed contextually relevant.

To better ascertain whether the STING signaling pathway is required for activation of adaptive immune responses *in vivo* in response to combination temozolomide and anti-CD47 therapy, we have repeated combination therapy in STING knockout animals (MPYS^{-/-}(STING^{-/-}) (B6(Cg)-*Tmem173*^{tm1.2Camb/J}). Immunofluorescence detection of phospho-IRF3 (STING) show that the combo treatment of TMZ and anti-CD47 strongly induced IRF3 phosphorylation of CD45⁺ cells in the WT mice but not in the KO mice, which confirmed the lack of STING signaling in this strain (**Fig. 4e-g**). Furthermore, we found that after the combo treatment, the total tumor infiltration by T lymphocytes detected by immunofluorescence staining was significantly reduced in STING knockout animals, and the treatment efficacy was less robust in those KO mice compared with the WT (**Fig. 4h,i**). We believe these results confirm the importance of STING signaling to activate immune *response in vivo*.

3) *In vivo* experiments should not just be conducted with GL261 as this is a highly criticized murine glioma model, with questionable immunologic relevance. Both mechanistic and survival experiments should be done side by side in the CT2A model, to permit comparisons of the data across multiple models.

We agree with the reviewer that it is important to recapitulate our findings in more than one tumor model. We performed temozolomide/anti-CD47 ab therapy using the CT-2A model, and the data is presented in **Supplemental Fig. 14**.

4) As a smaller concern, it is not clear that improving a phagocytosis index from 10% to 20% is altogether relevant. Although the authors do demonstrate improved antigen presentation (although again unfortunately only in BM APC), some demonstration that such an improvement is truly important might be in order. It is acknowledged however that that is likely a difficult task.

We thank the reviewer for their insightful question. We agree that the significance of incremental increases in phagocytosis may be difficult to infer as meaningful and clinically relevant. However, we do note that: 1) In tumors that express high levels of CD47, phagocytosis is largely inhibited and therefore we find that quantitative assessment of phagocytosis is a reliable method for determining response to anti-CD47

therapy. 2) Although anti-CD47 mAb induces phagocytosis, this process appears to be 'sterile' in the context of the glioblastoma models used in this study, as it fails to induce antigen presentation by macrophage cells (**Figure 3f**) and only a minimal gain in survival is observed GL261 tumor bearing mice treated (**Figure 1d**). 3) Temozolomide also promotes tumor cell phagocytosis by macrophage cells through a mechanism independent of CD47, but as a monotherapy it similarly fails to induce substantial immune activation both *in vitro* (**Figure 3f-i**) and *in vivo* (**Figure 5**). 4) The addition of anti-CD47 mAb to temozolomide therapy causes the shift towards adaptive immune activation, resulting in T cell mediated tumor cell killing as shown in **Figure 5h,i**. Therefore, the improvement in phagocytosis index albeit only 10% in absolute terms, appeared to produce significant therapeutic effect likely due to potential amplification of antitumour immune responses down-stream, hence the key notion that phagocytic induction is required to promote adaptive immune activation which is the main player in promoting ultimate tumour control.

Because there are qualitative differences in the type of phagocytosis that can occur: sterile (non-inflammatory) or inflammatory, measuring the ability of a given innate immune cell to present antigen in addition to phagocytic capacity may be more predictive of therapeutic response than measuring phagocytosis alone. We demonstrated this successfully with combination temozolomide and targeted CD47 blockade, where an increase in both phagocytosis and antigen presentation is observed *in vitro* (**Figure 3f**), and results in a significant survival advantage *in vivo* (**Figure 5a,b**). Therefore, improvement in phagocytosis index by itself will require coordinated assessment of other immune activation processes in the setting of combination treatment.

Reviewer #2 (Remarks to the Author): Expert in phagocytosis

This is an important study regarding a combination therapy of CD47 blockade and temozolomide. The authors revealed a mechanism of TMZ-induced CRT exposure on glioma cells which induced phagocytosis by macrophages and activation of APC functions of macrophages, and proposed a novel treatment approach for targeting glioma. The research topic is very important and the experiments were very well designed and performed.

Despite the enthusiasm, a few concerns need to be addressed by the authors, before the manuscript can be considered for a publication in Nature Communications.

1. Temozolomide is one of the standard malignant gliomas treatment chemotherapy drugs. TMZ treatment elicits apoptosis and autophagy of glioma cells as documented in previous studies. While an experiment was performed in the study (Fig.S6) to show TMZ didn't significantly induce apoptosis, additional experiments are needed to address

1.1) what are the differences regarding the doses and timing of TMZ treatment between this study and previous studies when TMZ induced apoptosis in glioma cells;

We would like to preface our response by first mentioning that we acknowledge that genetic drift and other selective pressures applied to passaged cell lines over time may contribute to differences observed in separate research studies conducted in other laboratories. That being said, if we directly compare the doses and timing of temozolomide used in a recent paper (*Int J Mol Sci* (2019) 20(7):1562) the authors of this study publish a similar finding that temozolomide fails to induce significant cell death in LN229 human glioblastoma cells until time points beyond 96 hours. In a separate paper (*Acta Pharmacologica Sinica* (2014) 35, 832-838) the authors show that U87 human glioblastoma cell lines treated for 48 hours with temozolomide are resistant to doses equal to or lower than 200 μ M. These findings are consistent with our observations that temozolomide treatment does not induce substantial apoptosis in glioblastoma cell lines at doses and at time we tested. To further illustrate this point we show in Supplemental Figure 7, that a 72 hour TMZ treatment of murine and human glioblastoma does not induce severe apoptotic cell death until doses exceed 500 μ M. An important distinction that we make in this manuscript is that calreticulin translocation and resulting phagocytic induction are events that occur prior to apoptosis, as these transpire at lower doses and earlier time points than are reported by us and others for temozolomide-induced cell death to occur. Additional supporting evidence of this is presented in Figure 2B where at 96 hours following temozolomide treatment calreticulin expression drops, and cell death begins to increase.

1.2) what would be the threshold when TMZ induces apoptosis in the glioma cells used in the manuscript;

To test the threshold for TMZ to induce apoptosis, we used ultra-high doses of TMZ (up to 800 μ M) to treat four cell lines used in the study: U251, LN229, CT2A, and GL261 (**Supplemental Fig. 7a**). After 72h of treatment, apoptosis was evaluated by AnnexinV staining. We found the two mouse cell lines (CT2A and GL261) were very resistant to the TMZ-induced apoptosis, which was less than 15% even when the TMZ concentration reached 800 μ M. The solubility of TMZ in DMSO is 38 mg/ml (195 mM), 800 μ M is almost the maximum *in vitro* dose to achieve when keeping DMSO below toxic level. For the two human cell lines (U251 and LN229), 800 μ M induced apoptosis of 20-30%. 500 μ M of TMZ induced ~15% apoptosis in the LN229 cell line. Considering the relatively high level baseline apoptosis of the U251 cell line, we conclude that the TMZ dose > 500 μ M is the threshold to induce significant apoptosis in the human glioma cell lines.

1.3) What would be the effects on cancer cell phagocytosis and APC function of macrophages when TMZ reaches the concentration to induce significant apoptosis;

Since the mouse glioma cells were very resistant to TMZ-induced apoptosis (**Supplemental Fig. 7a**), we tested the effect of ultra-high dose TMZ on human glioma cell lines and the human macrophage cell line THP-1. We found that 800 μ M treatment of TMZ for 72 hour induced significantly increased apoptosis (~30%) of the THP-1 (**Supplemental Fig. 7c**). We expect the normal cell function will be impacted when

apoptosis reaches this level, more importantly, such a high dose like this is not clinical-relevant (please also see our response to question 1.1 and 1.4).

1.4) the authors should comment how the concentrations of TMZ used in in vitro experiments were translated to the doses used in the in vivo treatment, and how the doses of TMZ used in this study are relevant to its clinical use;

Standard dosing of temozolomide in patients is 150-200mg/m² given for 5 days per cycle. An average adult weighing 70kg would receive 350-400mg, which equals 5-6mg/kg for a total of about 25-30mg/kg in one cycle. Because rodents have a higher metabolic rate, allometric scaling is applied to achieve the appropriate therapeutic dose of temozolomide in mice (*J Basic Clin Pharm* (2016); 7(2): 27-31). Therein, the animal equivalent dose calculation based on body surface area (assuming a mouse of 20g) is to multiply the human dose by a factor of 12.3. Therefore the animal equivalent standard human dose of temozolomide applied to a mouse is 61.5-73.8mg/kg given daily for 5 days for a total of 307.5-369 mg/kg total. In our metronomic arm we used 20mg/kg given for 3 days, which totals 60mg/kg. In the non-metronomic arm we give 80mg/kg for 3 days, which totals 240mg/kg. Cumulatively, this comes out to 300mg/kg for the total treatment administered. If we divide by the conversion factor 12.3 for human comparison, it equals 24.4 mg/kg and is on par with the total amount of drug given to a human patient in one standard cycle of treatment.

2. Antitumor activity of TMZ is dependent on MGMT expression. The authors tested the effects of MGMT overexpression in a cell line that does not endogenously express MGMT. It would be more convincing if the authors could examine MGMT expression in multiple lines used in the study and investigate if there is a correlation between endogenous MGMT expression, resistance to TMZ-induced CRT translocation and susceptibility to phagocytosis;

We thank the reviewer for this thoughtful suggestion. To test this, we have purchased the U138 glioblastoma cell line, which are reported to express endogenous MGMT (*Genes Dis* (2016) 3(3):198-210). We have repeated calreticulin translocation and phagocytosis assays using the same dosing and time regimen previously used in this study. We confirmed that U138 expressed high level of MGMT and found that the U138 cells had very little ER stress induced by TMZ treatment, indicated by calreticulin translocation and protein markers (p-eIF2 α and CHOP) (**Supplemental Fig. 8f,g**). Accordingly, TMZ treatment only slightly increased (~15%) U138 phagocytosis by THP-1, whereas the phagocytosis was 70% higher after U251 was treated (**Supplemental Fig. 8h**).

3. The authors should comment on why TMZ treatment doesn't induce CRT translocation in normal brain tissue cells;

Cancer cells, glioblastoma included, experience heightened levels of cellular stress that persist as a result of exacerbated transcription, protein production, metabolic demands, etc. that support their malignant state (*Immunity* (2013) 39(1): 74-88). Unlike normal

cells, this causes them to be hypersensitive to agents that compound cellular stress, and in fact is one of the founding principles behind chemotherapy. Therefore, elevated endoplasmic reticulum stress and subsequent calreticulin translocation is observed in cancer cells at lower doses than would be necessary to achieve the same result in normal cells (*Nat Rev Immunol* (2017) 17(2):97-111). Expansive calreticulin translocation in normal brain cells could lead to the activation of auto-immune processes, cytotoxicity, and even death. In our experimental models, inflammation is localized within and around the tumor, supporting that therapy induced immune activation is tumor-specific. Auto-immune activation is also not observed in patients treated with temololomide, further lending support that immune responses are tumor specific.

4. Does TMZ treatment have a direct effect on cell proliferation of the glioma cells?

We tracked the proliferation of the glioma cells (Human: U251, LN229; mouse: GL261, CT2A) by MTS cell proliferation assay. The cells were cultured with or without temozolomide treatment. We found 100 μ M and 300 μ M temozolomide significantly inhibited the proliferation of those cells after 72h (**Supplemental Fig. 3**)

We are very grateful for all the time and efforts of the editor and the reviewers, and we hope that we have addressed all the reviewers' concerns. All changes are highlighted in the tracking changes document. Here we resubmit our revised manuscript, figures and supplementary figures. We are looking forward to your reply. Thank you very much!

Best Regards,

Wen Jiang, M.D., Ph.D.

REVIEWERS' COMMENTS:

Reviewer #1 (Remarks to the Author):

The authors have done a substantial amount of work to address the concerns of this reviewer. I have no further comments or requests.

Reviewer #2 (Remarks to the Author):

My concerns have been adequately addressed.